# Femtosecond Laser Fabrication of Silver Microstructures in Nanoporous Glasses

A. O. Rybaltovsky [1,2], E. O. Epifanov [2], V. N. Sigaev [3], S. S. Fedotov [3], V. I. Yusupov [2] and N. V. Minaev [2,*]

1    D.V. Skobeltsyn Institute of Nuclear Physics, M.V. Lomonosov Moscow State University, Vorob'evy Ggory, 119991 Moscow, Russia; alex19422008@rambler.ru
2    Institute of Photon Technologies, Federal Scientific Research Centre 'Crystallography and Photonics', Russian Academy of Sciences, ul. Pionerskaya 2, Troitsk, 108840 Moscow, Russia; rammic0192@gmail.com (E.O.E.); iouss@yandex.ru (V.I.Y.)
3    Department of Glass and Glass-Ceramics, Mendeleev University of Chemical Technology of Russia, 125047 Moscow, Russia; sigaev.v.n@muctr.ru (V.N.S.); fedotov.s.s@muctr.ru (S.S.F.)
*    Correspondence: minaevn@gmail.com

**Abstract:** This paper presents the results of studying the process of laser formation of microstructures from silver nanoparticles in nanoporous quartz glasses. Glass samples were impregnated with organometallic molecules Ag(hfac)COD in a supercritical carbon dioxide environment. The formation of point and linear microstructures was carried out by high-frequency (70 MHz) femtosecond laser radiation with a wavelength of 525 nm and energy in the pulse up to 1 nJ. It was found that the formation of microstructures occurs due to photo- and thermal decomposition of precursor molecules with the formation of plasmonic silver nanoparticles. It is shown that the developed temperatures can exceed the melting point of glass, which leads to the appearance of microstructures with altered refractive index. A qualitative model explaining the individual stages of cluster formation in the glass volume under point laser impact is presented.

**Keywords:** femtosecond laser radiation; microstructures; nanoporous silica glasses; nanoparticles of silver; laser induced transformation; plasmonic properties





## 1. Introduction

Transparent glasses have been a topical subject related to the laser formation of bulk functional micro- and nanostructures over the past two decades [1–3]. In this subject area, a special place is occupied by technologies based on the use of the unique properties of plasmonic silver nanoparticles (Ag NPs) [1] formed in a glass matrix, for example, laser technologies for recording information with a high degree of density [2]; formation of waveguide and conductive structures for optoelectronics [4]; creation of sensitive sensor devices for the identification of gaseous and liquid impurities, including those for surface enhanced Raman spectroscopy (SERS) [5,6]; synthesis of materials with luminescent rare earth ions for use in laser systems [7]; and numerous applications in metamaterials, near field optics, microfluidics, and micromechanics [3].

Progress in some of these areas is mainly associated with the use of femtosecond laser radiation for the formation of bulk micro- and nanostructures. High-intensity radiation makes it possible to effectively decompose silver-containing molecules with the release of silver atoms and ions, followed by their self-assembly into nanoparticles [3]. The possibility of laser-induced decomposition in the regime of two-photon and multiphoton absorption allows the recording density of the formed structures in the matrix volume to be significantly increased. An increase in the density of microstructuring is associated in this case with a decrease in the effective size of a single microstructure. The possibility of forming functional structures using the plasmonic, fluorescent, and nonlinear optical properties of AgNPs and small clusters of silver atoms is of great interest [2]. One of the

most interesting demonstrations of this approach is the creation of devices for unlimited lifetime data storage with high-density capacity and architectures up to 6D [8,9].

It is important to underline that the choice of the type of glassy material as a carrier matrix for plasmonic nanoparticles has a significant effect on the final characteristics of the microdevices created. High-silica nanoporous glasses (NPGs) [10], whose synthesis was developed more than 40 years ago, are still the most promising for many applications due to their outstanding properties [2,3]. Thanks to their high transparency, such materials are excellent for information storage [8]. They are very convenient for the laser microstructuring process, making it possible to form submicron (of 100 nm order) structures [11] using femtosecond radiation. NPGs with functional nanoparticles distributed through the matrix volume present a great prospect for the fabrication of sensor systems due to the presence of a developed system of interconnected pores, as well as high chemical stability, transparency, and adsorption capacity [12]. The sensitivity of the entire system is ensured by a high degree of pore percolation, which in this case allows the detectable compounds to effectively circulate past the nanoparticles immobilized in the matrix volume. Also, the presence of nanopores in such glasses has a significant effect on the microstructures formation, which allows the implementation of the processes of ultrafast polarization-sensitive microstructuring by fs laser pulses [13]. Impregnation of the internal volume of the NPG matrix with various functional compounds (for example, organometallic compounds of silver or rare earth metals) is a simple and convenient method for bulk material functionalization by plasmonic nanoparticles and luminescent ions [7].

Earlier we [14–16] presented one of the possible ways to form AgNPs and their microstructures in the volume of NPG Vycor using continuous visible laser sources. In this case the silver precursor Ag(hfac)COD was introduced through impregnation of a porous matrix in a supercritical carbon dioxide ($scCO_2$) medium. Under the action of laser radiation quanta, photolytic decomposition of precursor molecules occurred creating $Ag^0$ atoms from which AgNPs were further formed. This synthesis method for a microstructured glass nanocomposite with AgNPs has some advantages if compared to the known technologies of multicomponent glass synthesis, where one of the components belongs to silver oxide [2]. One of the advantages is that the photochemical reaction products (organic residue molecules and unreacted precursor molecules) can be removed from the matrix volume by secondary extraction using $scCO_2$. Also, the Ag(hfac)COD precursor allows the use of low-intensity (the order of $10^2$–$10^3$ W/cm$^2$) continuous visible laser radiation to initiate the photoinitiated decomposition of Ag(hfac)COD [14], since it falls within the edge of the absorption band of the precursor [15]. The use of such a low intensity to form nanoparticle structures precludes the destruction of any bonds in the glass mesh elements, unlike in multicomponent glasses. The formation of AgNPs and structures from them in transparent polymer matrices has also occurred with the same photoinduced mechanism [17–20]. For example, ref [21] showed the possibility of the formation of gold nanoparticles and structures from them using tetrachloroaurate (H[AuCl$_4$]*4H$_2$O) as a precursor.

It is important to note that the silver precursor Ag(hfac)COD used here has a rather low decomposition temperature of ~120 °C. This allows us to obtain silver NPs in the matrix volume by heating to temperatures not exceeding 160 °C [16]. Earlier in our works we noted the interrelation of photo- and thermoinduced processes in the formation of similar nanoparticles [20] during the formation of AgNP structures in the polymer matrix volume.

During laser formation of microstructures from AgNPs in the volume of NPGs there are a number of features that are not yet fully understood. It is necessary to carry out a detailed study of the conditions under which the processes of transformation of light energy absorbed by AgNPs into thermal energy [22,23], which leads to the formation of new nanoparticles, occur in the irradiation zone. We considered this issue for the case of a polymer matrix [20]. However, in the case of porous glass matrices, it is also necessary to determine the influence of the porosity degree on the mechanisms of microstructure formation in the matrix volume. It is also necessary to study the influence of the process development dynamics in such structures under the action of fs laser pulses on the exposure

and focusing parameters. It is necessary to understand how the architectonics of the porous matrix itself changes in the irradiation zone.

The aim of the present work is to investigate the specific features of femtosecond-laser-forming of structures from silver nanoparticles in the NPGs impregnated with a Ag(hfac)COD precursor in s $scCO_2$ medium. The objectives of the study were to determine the main mechanisms of AgNP formation and to investigate the resulting microstructures under different focusing conditions and laser radiation doses.

## 2. Materials and Methods

Samples of nanoporous quartz glasses of two types were used as initial matrices (Table 1).

**Table 1.** Main characteristics of nanoporous glasses.

| The Main Characteristics of Glasses | Vycor | MCTU-RF |
|---|---|---|
| Specific density, $g/cm^3$ | 1.5 | 1.6 |
| Pore size, nm | 5 | 4.5–10 |
| Porosity, % | 28 | 20–25 |
| Refractive index | 1.33 | 1.39 |

The Vycor glass samples were fabricated using the method described in detail in [7]. The method is based on the process of leaching followed by washing out of the borosilicate phase from the common matrix of two-component glass. Samples of high silica quartz glass of the MCTU-RF type were fabricated at the Mendeleev University of Chemical Technology using a similar procedure to that described in [8]. Samples of both types of glass were presented as plates $5 \times 10 \times 1$ mm in size with polished edges.

The organometallic compound (1,5-ciclooctadiene) (1,1,1,5,5,5-hexafluoroacetylacetonate) silver(I)-Ag(hfac)COD was used as a silver precursor, where COD is cyclooctadienal ligand (CAS Number: 38892-25-0, Aldrich Chem. Corp., St. Louis, MO, USA). The initial glass samples were impregnated with the Ag(hfac)COD precursor in the $scCO_2$ medium, in which this silver complex dissolves well. For this purpose, the samples were placed in a 5 mL high pressure reactor together with 6 mg of the precursor wrapped in filter paper. The impregnation process was carried out according to the procedure described in our works [16,24] at temperatures of 40–50 °C for 1–1.5 h under 200 bar pressure.

Figure 1 shows a diagram of the setup used to study the formation of microstructures. A TEMA-100 femtosecond laser (Avesta Project, Russia) was used as a radiation source. Second-harmonic radiation with a wavelength $\lambda = 525$ nm, a pulse repetition rate of 70 MHz, a pulse duration of 200 fs, and an individual pulse energy of 0.95 nJ was fed into the lens through a system of mirrors. In this case, the average radiation power in the experiments could reach 200 mW. The sample was positioned in space using an ABL 1000 three-dimensional positioning system (Aeroteh, Pittsburgh, PA, USA) with submicron accuracy. To move the laser beam in the XY plane inside the working lens, a Hurry SCAN II 14 galvanic scanner (Scan Lab, Charles, IL, USA) was used.

Experiments on the microstructure formation were carried out with focusing of laser radiation both on the surface and in the bulk of the sample (Figure 1b). Also, two objectives with different numerical apertures were used (Table 2). The beam waist parameters were measured using an adapted scanning knife-edge method [25].

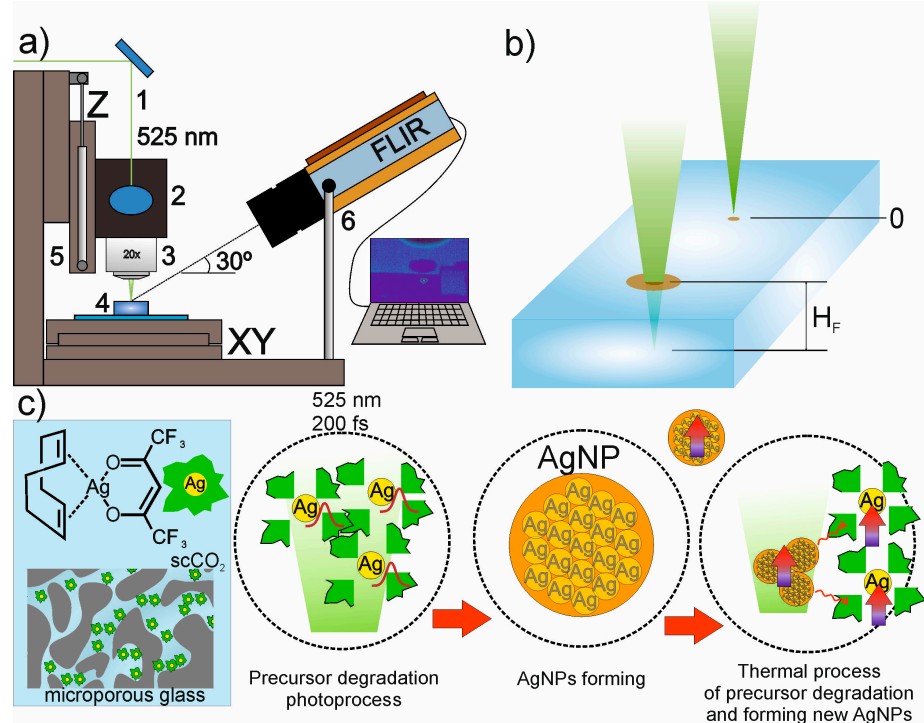

**Figure 1.** Scheme of the setup and the main stages of the processes leading to the formation of AgNP microstructures in impregnated porous glasses. (**a**) Scheme of the installation: 1—femtosecond laser radiation with λ = 525 nm; 2—galvanoscanner; 3—microscopic lens; 4—nanoporous glass sample; 5—three-axis motorized sliding motion; 6—thermal imager. (**b**) Two schemes of laser radiation focusing on the sample surface and 150 μm deep from the surface. (**c**) Scheme of the process of silver nanoparticle formation in porous glasses under the influence of laser radiation. Porous glasses are initially saturated with silver precursor in $scCO_2$ medium. Then its photodegradation with Ag0 formation and the assembly of AgNPs takes place. Under further laser exposure, AgNP aggregates are heated, accelerating the precursor decomposition process.

**Table 2.** Specifications of objectives for laser beam focusing.

|  | Lens Type | Work Distance, mm | Beam Waist Radius $\omega_0$, μm | Beam Waist Length ZR, μm |
|---|---|---|---|---|
| 1 | Optosigma PAL-20-L, S Plan Apo HL 20×/0.29 | 31.1 | 1.7 | 17.3 |
| 2 | ZEISS 422050-9903-000, EC EPI PLAN 20×/0.4 | 3.2 | 1.1 | 7.2 |

To visually control the formation of microstructures, a digital camera (ToupTek XF-CAM1080PHB, Hangzhou, China) with a telescopic lens and manual focus adjustment was used. Through this camera we managed to obtain an image of 2 × 3 mm in the area of irradiation in real time. For better visualization, a red LED illumination was employed in transmitted light in bottom-up geometry.

Registration and identification of the formed AgNPs on the surface and in the bulk of the glass were carried out using the plasmon absorption spectra [26,27]. For this purpose, regions sized 400 × 400 μm, consisting of linear tracks, with a fill distance of 5 μm, were created in the samples with the help of laser radiation. As a result, a square-shaped region was formed on the surface, the color of which corresponded to the plasmon absorption of AgNPs. The absorption spectra were recorded using an Ocean QE Pro fiber spectrophotometer (Ocean Insight, Orlando, FL, USA) with frontal illumination from a

stabilized source of broadband radiation in the visible and UV spectral region DH-2000 (Ocean Insight, Orlando, FL, USA).

In this work, the temperature distribution dynamics on the surface of a porous glass sample was recorded using a FLIR A655sc thermal imager (FLIR Systems, Wilsonville, OR, USA) with an additional Close-up IR Lens 5.8×, which helped to obtain a spatial resolution of about 100 µm/pixel. The thermal imaging camera was installed at an angle of 30° to the horizontal plane at a distance of ~20 cm from the point of intersection of the optical axis with the sample surface (Figure 1). The registration of thermograms was carried out up to a temperature of 160 °C with shooting frequency of 50 frames/s. In this case, thermography was carried out only when using a long-focus lens. The obtained data were processed with the FLIR tools software.

Images of plasmonic microstructures in irradiated samples were recorded with an HRM-300 Series 3D microscope (Huvitz, Anyang, Republic of Korea) equipped with a U3CMOS05100KPA digital camera (Touptek, Singapore). Digital image processing of AgNP structures made it possible to analyze the features of the bulk formation of nanoparticles in NPG samples. It was assumed that the optical density of the sample area, $OD = \log_{10}(I_0/I)$, where $I_0$ and $I$ are the intensities of the incident and transmitted light, respectively, is proportional to the thickness of the structure and the concentration of formed nanoparticles in it [28]. To estimate the concentration of AgNPs from an optical image obtained in transmitted light, the dependence for light attenuation $I = I_0 \exp(-A \cdot d \cdot C)$ was used, where $I$ is the intensity of the transmitted light, $I_0$ is the intensity of the incident light, $A$ is a coefficient depending on the features of scattering and absorption of light nanoparticles, $d$ is the structure diameter, and $C$ is the concentration of nanoparticles. The concentration was estimated under the assumption $A = const$ by the expression $\exp(-C) = \ln(I_0/I)/(A \cdot d)$.

The calculation of the temperature under point laser action on the surfaces of porous glass samples was carried out in accordance with [29]. The coefficient of thermal diffusivity for nanoporous glass was assumed to be $a = 10^{-6}$ m$^2$/s.

## 3. Results

### 3.1. Laser Formation of Point and Line Structures on a Surface

Figures 2 and 3 show examples of dot and line microstructures (array of dots) and (array of lines) on glass surfaces, which were formed using a 20X short throw lens with NA 0.4 (no. 2 of Table 2), which gives an opportunity to form structures with minimal transverse dimensions. The resulting microstructures have a brown color, which is characteristic of AgNP cluster formation with plasmon absorption bands in the region above 400 nm [26].

In Vycor glass, dot structures are clearly observed at an average power of $P = 10$ mW, while, in MCTU-RF glass, even at $P = 20$ mW only a faint spot is visible (Figure 2a). As the power increases, the spot diameter increases, which is confirmed by the analysis of the optical density distribution from their optical micrographs (Figure 2a–c). At the same time, the density distribution in the spot has a rather complex concentric structure with a dark core in the central region, which is explained by an increased concentration of AgNPs (Figure 2b).

Figure 2c presents the results of the dependence analysis of the spot diameters D on the power P at a constant exposure t. It can be seen that, for both samples, the dependences $D(P)$, and hence $D(F)$, where $F = P \cdot t$ is the fluence, are well approximated by straight lines.

It is known that, in the classical case for Gaussian beams in the presence of the structure formation threshold $F_{th} = const$, a linear dependence exists between the square of the structure diameter and $\ln(F)$ [30]. Therefore, it can be argued that, in our case, the AgNP formation threshold is not const, but depends on the value of the laser radiation fluence. Since the spot diameter of AgNPs increases faster than in the classical case, it follows that the threshold for their formation decreases with increasing fluence. Such a phenomenon can occur due to the fact that the process of thermolysis of precursor molecules due to an increase in temperature in the area of influence is added to the process of photolysis of precursor molecules under the action of laser radiation. Such an increase in temperature

under the action of laser radiation occurs due to the effective transformation of the absorbed light energy into thermal energy by silver nanoparticles and aggregates that have plasmon resonance [22,23].

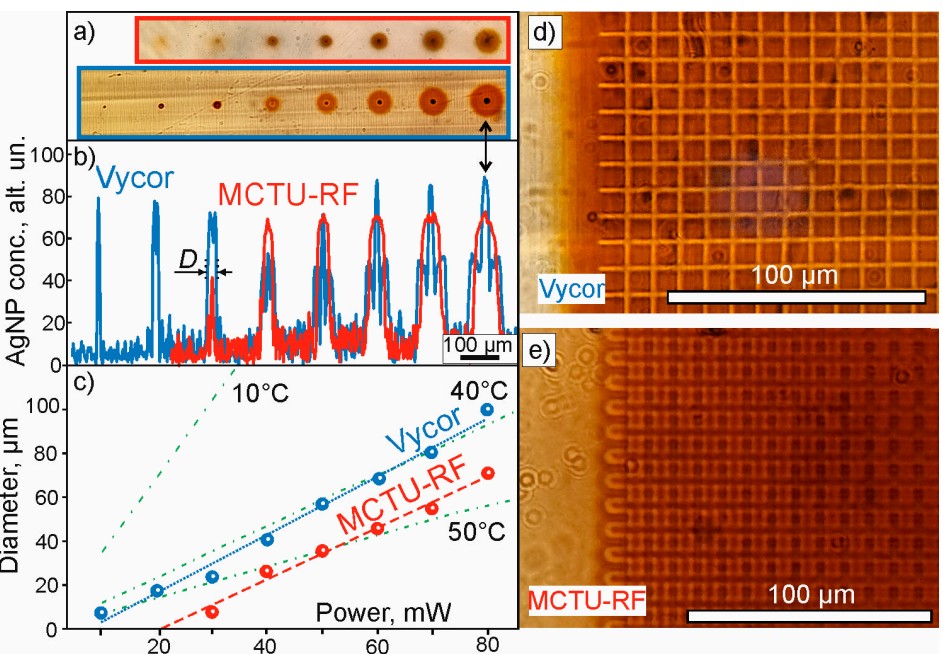

**Figure 2.** Microstructures on the surfaces of Vycor and MCTU-RF glasses. (**a**)—Photographs of the point structures obtained on the Vycor sample (blue rectangle) and MCTU-RF (red rectangle) by varying the laser radiation power. The distance between the points is 150 μm. (**b**)—The corresponding distribution profiles of AgNP concentrations for point structures: blue curves—for the Vycor sample, red curves—for the MCTU-RF sample. (**c**)—Dependence of the outer diameter of the formed point structures on the radiation power at exposure time at each point t = 0.5 s. Dash–dotted lines labeled "10 °C", "40 °C", and "50 °C" show the positions of the isolines of temperature rise to 10 °C, 40 °C, and 50 °C, respectively, calculated assuming a spot heating model. (**d,e**)—Photographs of lattice structures. Beam travel speed = 50 μm/s, P = 20 mW. Lens no. 2 was used (Table 2).

Figure 2 with dash–dotted lines shows the positions of the isolines of the temperature rise on the sample surface, calculated under the assumption of a point heating model. It can be seen that the obtained trends for the diameters of point structures fit well on the isoline of 40 °C for the Vycor and 50 °C for MCTU-RF samples. Thus, the hypothesis of a faster growth in the diameters of microstructures with increasing laser radiation power compared to the classical case, associated with the synergistic effect of laser radiation and heating, can be considered quite reasonable. The results obtained in Figure 2c made it possible to determine the limit of this synergistic effect by heating, which is 40 °C for Vycor porous glass samples and 50 °C for MCTU-RF samples. The higher limit for the RF samples compared to Vycor, we believe, is associated with a lower concentration of the precursor in it.

Figure 3 shows examples of linear and dot structures on the surface of Vycor glass before and after heat treatment at 600 °C. During the formation of the structures, the laser beam was focused on the surface of the large face A of the sample (Figure 3d). It is clearly seen (Figure 3d) that the linear track formed along the trajectory leaving face B has a three-dimensional profile. The profile depth is ~10 μm deep and ~5 μm wide. The resulting structures have a pronounced brown color (Figure 3a,e), which disappears after the heat treatment of the sample in air at 600 °C (Figure 3b,f). It should be noted that this temperature was chosen experimentally and is optimal for erasing structures from AgNPs. As our studies have shown, at temperatures above 600 °C, gradual degradation and nanopore collapse can occur in the structure of Vycor glass. However, in our case, after the heat treatment, the structures, having significantly weakened their brown color,

were still preserved. It can be assumed that, as a result of strong heating of the glass matrix in the region of laser radiation focusing, the degradation of Vycor nanopores occurs simultaneously with the formation of AgNPs. Therefore, after annealing, a microstructure of degraded nanopores remains intact in the glass.

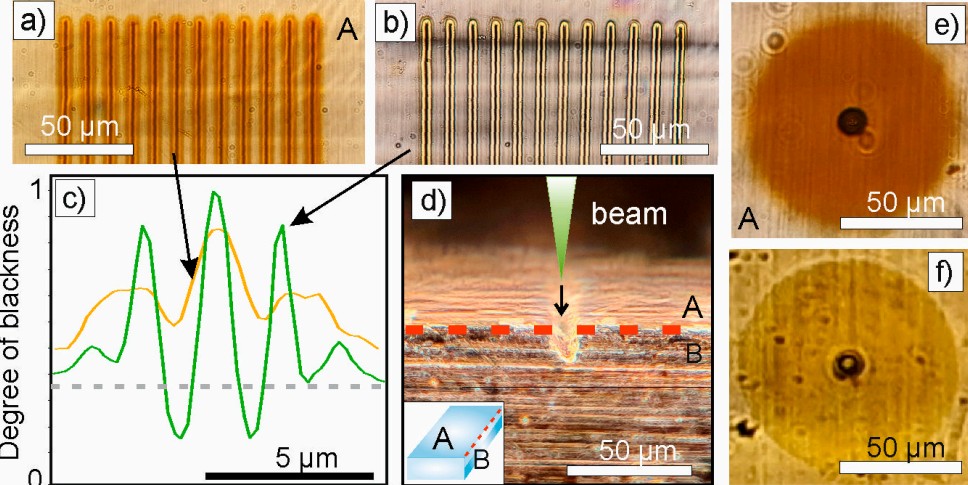

**Figure 3.** Optical photographs of linear and point structures on the surface of Vycor glass before and after heat treatment at 600 °C: (**a**)—Initial linear structures before heat treatment. (**b**)—Linear structures after heat treatment. (**c**)—Comparison of the cross-section of linear structures before and after annealing (pixel blackness in the cross-section of the structures is shown). (**d**)—Linear structure (side view, a red dotted line marks the edge between faces A and B). (**e**)—Initial point structure before heat treatment (P = 80 mW). (**f**)—Dot structure after heat treatment. When forming linear structures, the speed of the laser beam was 50 μm/s, P = 30 mW.

Note that the linear structures formed from AgNPs have a complex color distribution profile on the surface along their width (Figure 3a), which changes significantly in the case of heat treatment of the samples (Figure 3b). Detailed ideas about the changes in the color density of the track along its width are given by those results of its digital processing shown in Figure 3c, that are presented for both cases. The curve (yellow), corresponding to the original structure before heat treatment, has a central large peak and two peaks of lower intensity to the left and to the right of it. The curve (green) of the heat-treated sample differs significantly from the initial one: with an increase in contrast, all observed peaks are clearly narrowed, as a result of which the side peaks are obviously divided into two more. It can be assumed that the appearance of such side maxima is due to the formation of AgNPs in the region of scattered laser radiation from the resulting dense accumulation of nanoparticles along the axial line of the track. Similar phenomena can also explain the appearance of concentric rings around the central denser spots in the case of point laser irradiation (Figures 2a and 3e,f).

The increased contrast of the observed peaks (Figure 3c), associated with their narrowing during annealing, is explained by the movement of silver atoms and the formation of more compact agglomerations from AgNPs in the regions where they already existed. Such transformations in their distribution may well take place, given the following point. At the heat treatment temperatures used (~600 °C), the initial nanoparticles (less than 4 nm in size) begin to change their appearance due to their melting and evaporation [31]. In this case, it becomes possible to move silver atoms through the pores in the glass with the formation of new larger thermally stable nanoparticles. It can be assumed that the condensation of new nanoparticles with a higher melting point will occur in a limited central region. Here, the probability of deformation of the initial pores with a tendency to increase their size under the action of femtosecond radiation is the highest at the maximum beam intensity. Note that the well-known formation of pores [32] in ordinary pure quartz glasses under

the action of fs pulses occurs according to a different mechanism and is associated with a very high pulsed pressure during the laser breakdown of glass. The formation of void microchannels in nanoporous glasses also occurs according to another mechanism [33]. Here, voids formed in the glass matrix, preliminarily saturated with water, due to the action of high impulse pressures on the material heated to high temperatures. In our case though, the impact of femtosecond radiation from the viewpoint of pore transformation can be more effective because of the photothermal stimulation processes of AgNPs.

As for the observation of a faintly expressed outer contour for each track in Figure 3b, it corresponds to the most extreme peaks on the green curve in Figure 3c. Physically, this contour probably corresponds to the boundary of the region with a changed refractive index of porous glass. Such a change may well happen due to the deformation and destruction of the pore walls in the intime environment of the irradiation point during the development of thermal processes on AgNPs. Note that a certain boundary is also visible during annealing of point structures (see Figure 3f).

Thus, it can be argued that we have managed to create heat-resistant (within temperatures up to 600 °C) micro- and nanostructures in porous quartz glasses using ultra-low energies (of the order of 1 nJ) of femtosecond laser radiation.

Figure 4 shows the absorption spectra of the structures on the surface of the Vycor sample formed at different laser radiation powers of 10, 20, and 30 mW. It can be seen that prior to exposure Vycor glass impregnated with a silver precursor has an absorption band in the ultraviolet region [14,16]. When the samples are irradiated, an intense absorption band of AgNPs appears with a maximum in the region of 410 nm, associated with the presence of plasmon resonance. As P increases, the absorption gradually increases and the maximum of the band barely shifts. This indicates the formation of spherical AgNPs about 4 nm in size in glass pores [27]. As noted in [14,15], nanoparticles of this type also appear in impregnated Vycor glasses subjected to the action of continuous laser radiation with a wavelength of 532 nm. Based on the shape of the plasmon absorption spectra (Figure 4) it can be assumed that the shape and size of the synthesized AgNPs in glass pores remain almost unchanged at the given radiation parameters.

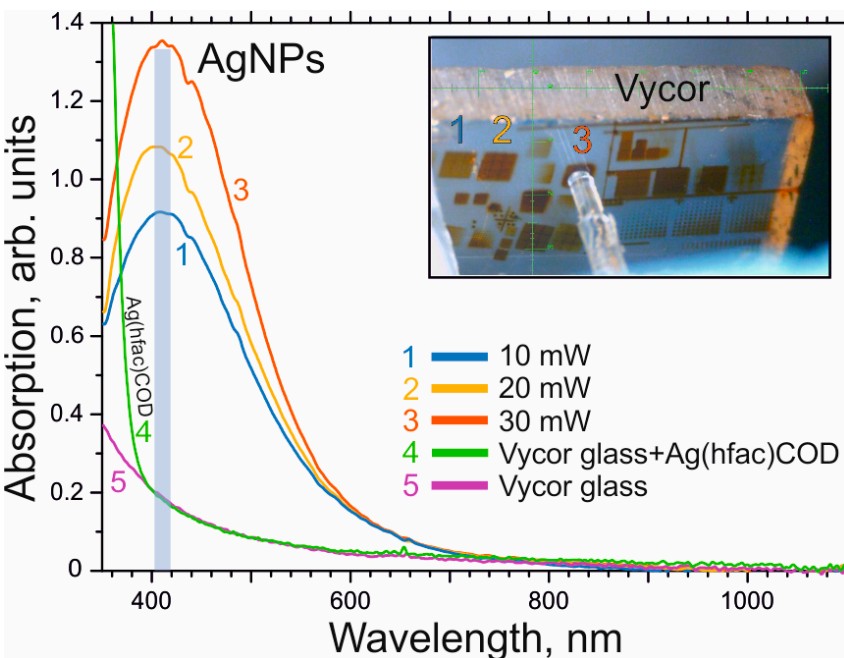

**Figure 4.** Absorption spectra of AgNP structures formed in Vycor glasses at different values of the average power at a laser spot velocity of 50 μm/s. The inset shows a photograph of a Vycor glass sample with surface structures. The purple spectrum (5) represents the pure Vycor glass without precursor and prior to exposure. The end face of the optical fiber of the spectrophotometer is attached to structure 3.

### 3.2. Laser Formation of Clusters from AgNPs inside the Volume of Glasses

Let us consider the specific features of the formation of clusters of AgNPs in the bulk of a matrix under the action of a point effect of femtosecond laser radiation into the bulk of porous glass. During the experiments, the laser beam was focused on a narrow side face B $2 \times 10$ mm in size (Figure 5a,b). Due to the fact that face A of the Vycor glass sample was transparent, the process of structure formation of the AgNPs was observed through it with the help of an optical microscope.

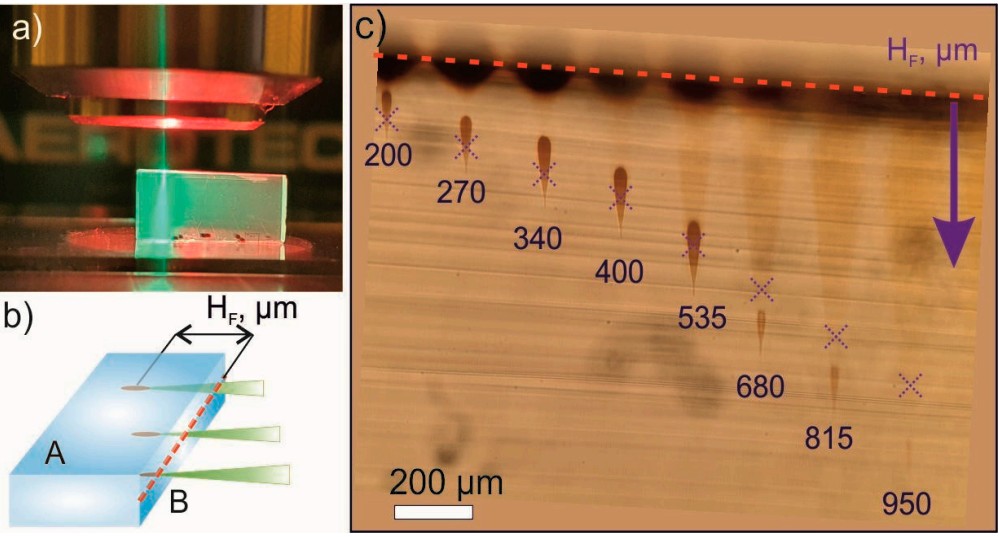

**Figure 5.** AgNP drop-shaped structures in Vycor glass, obtained by focusing laser radiation at different depths from the surface of the "B" facet. (**a**) Photograph of the specimen under the lens during irradiation. (**b**) Scheme of the experiment. The surfaces of faces "A" and "B" are shown. (**c**) Photographs of the resulting structures in transmitted light. The numbers show the depth of the structure $H_F$ from the surface of face B (shown by the red dotted line). Dotted crosses mark the places of focusing in air. P = 40 mW, t = 20 s.

Figure 5c shows transmitted light photographs of the formed drop-shaped structures. It can be seen that, as the $H_F$ focusing depth increases, the dimensions and contrast of these structures gradually decrease. As can be seen, the formed microstructures in the sample are somewhat deeper than the set values of focusing in air, which is explained by the higher refractive index of glass compared to air. It should be noted that similar drop-like structures were obtained earlier in [4] upon focusing femtosecond radiation in samples of multicomponent phosphate glasses doped with silver. However, in our case, the intensity was used two orders of magnitude lower.

Besides the above-described drop-like structures in the zone of the laser beam impact, the appearance of larger volumetric structures in the form of inverted cones is observed when exposed to greater depths. At the same time, drop-shaped structures are located in the region of their vertices. Based on their brown color, these cone-shaped structures can be characterized as accumulations of AgNPs (Figure 5c) that originate in the zone of the laser beam passage. Note that darker diffuse regions with an increased content of nanoparticles are formed at the base of the cones. This effect, in our opinion, is caused by a rather high concentration of defect centers of various natures in the near-surface layer of Vycor glasses [34]. A larger number of precursor molecules can be localized on such centers than in the rest of the volume. One reason for this is that this situation may occur due to the damage to the near-surface layer of porous glass during the polishing process of the sample or through some other processing.

Let us note that near-surface structures of noble metal nanoparticles, similar in nature, were observed by us earlier in polymeric materials [35]. As can be seen from Figure 5 the linear dimensions of such structures are more than 150 μm for nanoporous glasses. We

believe that not only the processes of photolytic decomposition of precursor molecules but also thermal processes are actively involved in their synthesis. They are known to increase not only the rate of precursor decomposition but also the rate of self-assembly of AgNPs. Obviously, large near-surface structures of AgNPs can partially prevent the formation of drop-like structures deep in the sample due to the screening effect.

Combining all the reasons listed above, the dimensions of the resulting drop-shaped structures initially slightly increase with increasing focusing depth. Thus, at a depth of 200 and 270 µm these structures are smaller in size than similar structures at depths of 340, 400, and 535 µm (Figure 5c). We believe that this effect is mainly associated with a gradual decrease in the radiation screening by near-surface structures due to a decrease in the laser fluence in this region. Then, as the depth increases (535 µm → 950 µm), the size of teardrop structures rapidly drops. Although the screening effect gradually decreases in this case, such a decrease in the size of the structures can be explained by a significant gradual decrease in the concentration of the precursor with depth.

For a more detailed study of the drop-like structure formation features of AgNPs in nanoporous glasses, experiments were carried out with porous glasses of different porosities (Vycor and MCTU-R). In these experiments, laser radiation was focused to a depth of $H_F$ = 250 µm with a power of P = 80 mW and the exposure time t was varied.

Figure 6 shows that the teardrop structures in Vycor and MCTU-RF glasses are visibly very different. In Vycor glass, they are more contrasting, wider in maximum diameter, and appear at a lower t than in MCTU-RF. In this case, the main difference between the structures in Vycor glass is the absence of the lower part of the structure in the form of a cone below the focusing plane. It can also be seen that the structures of AgNPs in MCTU-RF glass are much closer to the shape of the laser beam in the beam waist region.

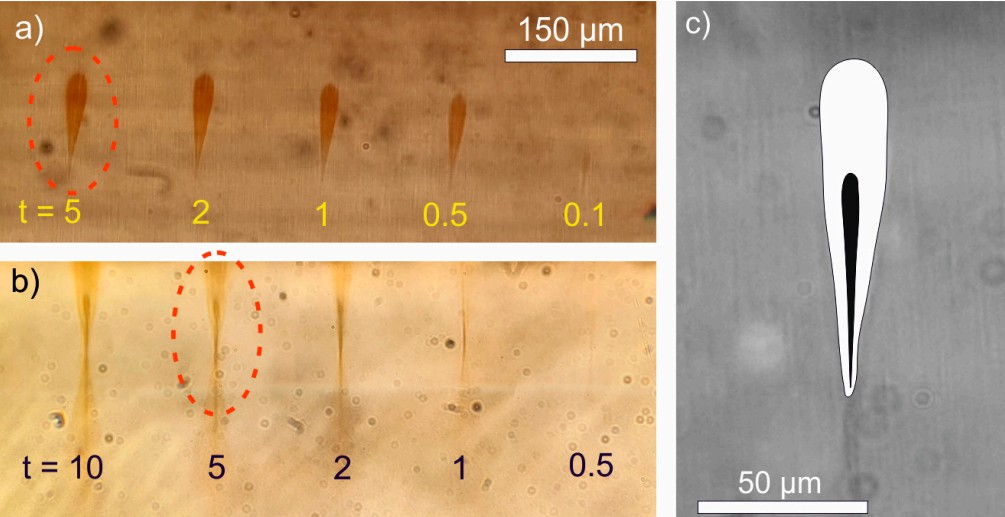

**Figure 6.** Optical photographs in transmitted light of AgNP drop-shaped structures in Vycor (**a**) and MCTU-RF (**b**) glasses at different exposure times t. The numbers indicate t in seconds. (**c**) Superposition of the contours of two structures (circled with a red dotted line in (**a**,**b**)) at t = 5 s, P = 80 mW.

The observed transition from needle-like to tear-shaped structures in Vycor glass is explained by the active influence of photothermal processes in the formation of AgNPs when the irradiation exposure changes from 0.1 to 0.5 s. The thing is that, with longer exposure, accumulation of AgNPs occurs not only in the constriction zone but also a bit higher. Under these conditions, an avalanche-like process of nanoparticle formation begins to develop, which is determined in space by the laser beam profile.

### 3.3. Photothermal Effects in Laser Drawing of Structures

The laser-induced formation of structures of AgNPs on the surface and in the volume of porous glasses impregnated with the precursor is accompanied by heat release. This is linked to the fact that the laser radiation wavelength falls within the intense plasmon absorption band of AgNPs [19]. Thermal imaging video recording makes it possible to record the dynamics of changes in thermal fields on the surface of samples during laser exposure (Figure 1a). Figure 7 shows the dynamics of the change in the maximum temperature on the surface of the Vycor and MCTU-RF glass samples during the formation of point structures on their surface and in their volume.

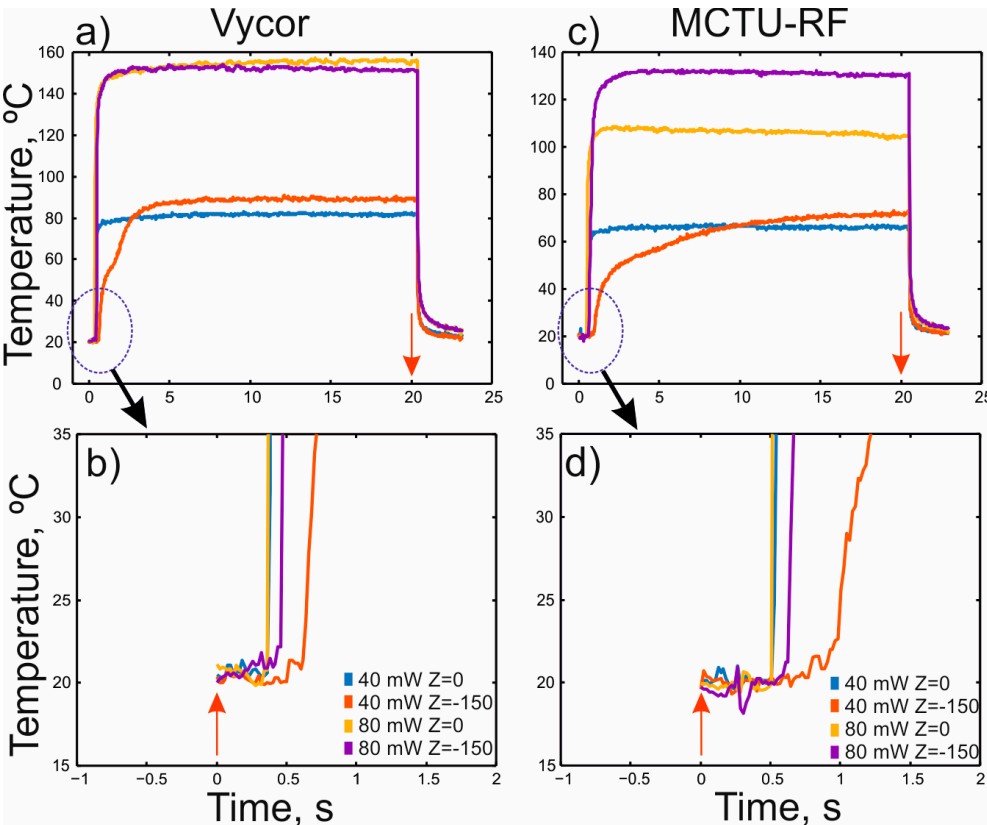

**Figure 7.** Dynamics of the maximum temperature on the surface of porous glass samples as a function of the time of exposure to laser radiation. (**a**,**b**)—Vycor glass and (**c**,**d**) MCTU-RF. In (**b**,**d**) the initial parts of the heating curves are shown on an enlarged scale. The red arrows mark the laser on time (↑) and the laser off time (↓).

Figure 7a,c shows that, some time after the laser is turned on, the maximum temperature on the surface of the samples rapidly increases and quickly reaches a plateau, and, after the laser is turned off, it also quickly decreases. In this case, a rapid increase in temperature on the sample surface is recorded after the start of irradiation only after 0.3–0.6 s on the Vycor sample and after 0.5–1.0 s on the MCTU-RF sample. At lower powers and with laser focusing inside the volume, the time interval before the start of the temperature increase grows (Figure 7b,d). For both samples, the slowest multistage temperature increase on the sample surface is observed when radiation is focused deep inside the matrix at P = 40 mW (Figure 7a,c).

As expected, the maximum surface temperature during sample irradiation is proportional to the laser power used (Table 3). It should be mentioned that, for the same radiation parameters, the recorded maximum temperatures for Vycor are higher than for MCTU-RF.

**Table 3.** Maximum temperatures (°C) during laser irradiation (Figure 7a,c) for different glasses.

| Power, mW | Temperature for Vycor, °C | | Temperature for MCTU-RF, °C | |
|---|---|---|---|---|
| | $H_F = 0$ | $H_F = 150~\mu m$ | $H_F = 0$ | $H_F = 150~\mu m$ |
| 40 | 62 | 70 | 47 | 52 |
| 80 | 136 | 132 | 88 | 112 |

We believe that the observed delay in the stepwise increase in temperature on the surface of the samples (Figure 7b,d) is associated with the formation of nanoparticles in the irradiated region, after which the avalanche-like process of the AgNP structure growth starts to develop. Taking into account that the concentration of precursor molecules in the near-surface layer is higher than in the bulk, the process of heating AgNP structures begins earlier upon focusing on the sample surface.

The two-step shape of the curves describing the temperature dynamics at $H_F = 150~\mu m$ and P = 40 mW is explained by the delayed appearance of an additional bulk heat source (large near-surface structures of AgNPs (Figure 5c)). The diameter of such a structure on the sample surface approximately corresponds to the diameter of the region where the laser beam enters the sample, focused at a depth of 150 μm. Since the radiation intensity in this case on the surface is lower than in the beam waist region, the rate of photolytic decay of precursor molecules is also lower. At the same time, the concentration of precursor molecules decreases with depth of focus. Both of these phenomena can affect the rate of the surface spot formation (Figure 5c) and the temperature rise on the sample surface (Figure 7a,c).

Let us estimate the time delay Δt between the switching on of laser radiation and the region with a rapid increase in the temperature of the sample surface in the case when the temperature source appears at a depth of 150 μm. Taking into account the refractive index n~1.33, the true focusing depth was ~200 μm. Evaluating the thermal diffusivity as a = 0.009 cm$^2$/s [36], we obtain $\Delta t \sim H_F^2/a \sim 44$ ms. This estimate shows that the observed large delays ~500–1000 ms cannot be explained by this effect (Figure 7b,d).

*3.4. Model of the Mechanism of Formation of AgNP Structures*

We propose the following mechanism for the AgNP structure formation on the surface and in the volume of matrices, which explains the dynamics of temperature growth (Figure 7). Initially, photodecomposition of the precursor with the release of Ag0 atoms and their self-assembly into AgNPs occurs in the region of laser exposure (Figure 1b). Appearing nanoparticles and their aggregates are intensively absorbing laser radiation and become heat sources themselves. Because of this, the temperature in the region of the formed nanoparticles increases. As a result, a synergetic effect occurs, when thermolysis is added to photolysis, and because of the increased temperature the diffusion rate increases, which leads to faster self-assembly of AgNPs and their aggregates. Thus, there is an avalanche-like acceleration of the processes, which leads to a jump-like temperature rise (Figure 7). The limitation of this growth, i.e., plateauing (Figure 7a,c), is related to the establishment of equilibrium conditions with the inflow of laser energy and its dissipation through thermal processes into the air and into the sample volume.

Fixation of lower values of maximum temperatures for MCTU-RF, in comparison to Vycor (Figure 7a,c and Table 3), at the same exposure parameters can be explained by a smaller specific internal pore surface in MCTU-RF glasses. This is the reason for a lower concentration of AgNPs, which finally leads to a lower heat dissipation under laser exposure.

Let us estimate the temperature that occurs during laser formation of structures on the surface of the material. When exposed to a series of pulses of focused laser radiation, two characteristic cases can be considered:

$$(1)~\sqrt{at} \ll \omega_0 \text{ and } (2)~\sqrt{at} \gg \omega_0, \tag{1}$$

where $a$ is the thermal diffusivity of the matrix, t is the observation time after the pulse, and $\omega_0$ is the waist radius for lens no. 2 (Table 2).

The first case is valid during the action of one femtosecond pulse ($\tau$ = 200 fs, pulse duration) since $\sqrt{a\tau}$ = 0.45 nm $\ll \omega_0$ = 1.1 μm. With this ratio, by the end of the laser pulse, the temperature will increase by [37]

$$\Delta T_p = \frac{2AP_p\sqrt{a\tau}}{\sqrt{\pi}\omega_0{}^2 k},$$

(2)

where A is the absorbing capacity of the AgNP structure, $P_p = \frac{P}{\tau f}$ is the average power per pulse, f is the pulse frequency, and k is the thermal conductivity of the material. In expression (2), all quantities are known, except for A.

Let us estimate A using the experimentally determined values of the temperature jump $\Delta T$ in the region of the optical axis for long observation times (second case). At $\sqrt{at} \gg \omega_0$ or t $\gg$ 17 ms, thermal equilibrium is established in the system. In this case, the temperature jump at the center of the laser spot on the surface is:

$$\Delta T = \frac{AP}{\pi\omega_0 k}.$$

(3)

Knowing the temperature jump $\Delta T$ from (3), we can find the desired absorption capacity A.

When evaluating the maximum temperatures that occur during laser irradiation, it should be taken into account that the thermal imager gives integral temperature values from one pixel-sized areas (100 × 100 μm). Taking into account the angle of 30° at which the thermal imager is installed on the sample surface (Figure 1), such an elementary area on the sample surface will be 100 × 200 μm in size. Since these values are much larger than the size of the laser beam waist (Table 2), it should be expected that the temperature values recorded by the thermal imager can be significantly lower than the actual maximum values.

To estimate the value of $\Delta T$ in expression (3), we suggest using the temperature jumps obtained experimentally using thermal imaging measurements (Figure 7). At the same time, it is necessary to show how, by the end of the observation time (t~20 s), the temperature changes within one pixel (100 × 100 μm) on the thermogram. We will assume that the distribution of the temperature jump on the surface in the stationary case is described by the known relation for a point action with a constant power:

$$\Delta T = \frac{AP}{4\pi kR} erfc\left(\frac{R}{2\sqrt{at}}\right)$$

(4)

where R is the distance from the optical axis. Integrating expression (4) over the size of one pixel (100 × 200 μm), we obtain that the average temperature jump over a pixel located in the region of the optical axis will be only ~4% of the average temperature jump over a spot with R = 5 μm, where R is a characteristic the size of the formed teardrop structures (Figure 5).

Thermographic measurements (Figure 7 and Table 3) showed that for P = 80 mW the rise in surface temperature at the center of the spot was 136 °C for Vycor glass and 88 °C for MCTU-RF glass. According to the above estimate, in a spot with R = 5 μm for Vycor glass $\Delta T$~3400 °C, and for MCTU-RF glass $\Delta T$~2200 °C, which exceeds the melting point of quartz (1710 °C). Using these estimates of $\Delta T$ and relation (3), we obtain that the desired absorbance A of the AgNP structure for Vycor glass is A~0.3, and for MCTU-RF glass A~0.2. Such a difference in the absorbing capacity of the structures in these porous matrices can be associated with their difference in geometric dimensions (Figures 5 and 6) as well as with the average concentration of the resulting nanoparticles per unit volume. The achievement of temperatures above a thousand degrees is confirmed by the appearance of microstructures, which represent defects in the glass matrix (Figure 3).

### 3.5. Peculiarities of the Formation of Drop-Shaped Structures from AgNPs

Figure 8a shows the result of mathematical modeling of the laser radiation intensity distribution in the focusing region superimposed on the optical photograph of the AgNP structure formed in the volume of Vycor glass (Figure 5c, structure at a depth of 535 μm). Superposition of the area in which the intensity of laser radiation exceeds 30% of the maximum value on the image of the contour of the drop-shaped structure shows that their transverse dimensions are comparable in the lower part. In this case, with a decrease in the depth of the structure, its diameter rapidly increases, in contrast to the transverse size of the presented isoline of the laser intensity.

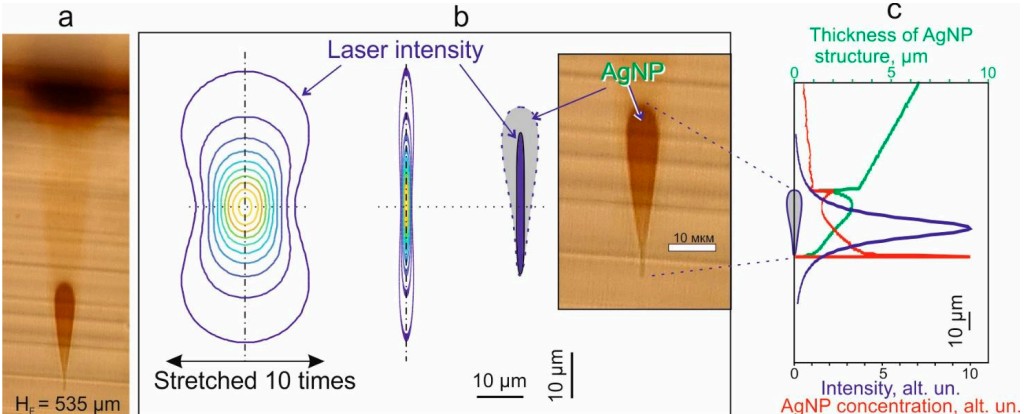

**Figure 8.** Comparison of the laser radiation intensity distribution in the focusing region with a photograph of the AgNP structure in Vycor glass. (**a**) Photograph of the analyzed structure in transmitted light. (**b**) Comparison of the laser intensity distribution with the contour of the formed structure (shown in Figures 5c and 8a). The area bounded by the isoline, inside which the laser radiation intensity is ≥30% of the maximum value, is colored blue. (**c**) Comparison of the change in the diameter of the AgNP structure with depth (green curve) with the intensity of laser radiation on the optical axis (purple curve) and the change in the calculated concentration of AgNPs with depth (red curve). The contour of the formed drop-shaped structure is also shown here.

Figure 8c shows the change in the diameter of the AgNP structure with depth as a green curve. Above the teardrop structure, the optical image shows a slight darkening shaped as an inverted cone (see Figures 5c and 8a), the change in diameter of which is reflected as a sloping area on the green curve (Figure 8c). In our opinion, such darkening is associated with the formation of nanoparticles in the volume of the sample, which is captured by the light beam until the moment of focusing. On Figure 8c, the red curve shows the calculated depth distribution of the AgNP concentration. Since the diameters of the drop-shaped structure formed in its upper and lower parts are minimal, the resulting darkening in these areas is caused by a very high concentration of AgNPs, which is reflected in the form of two "outliers" (horizontal sections) on the red curve (Figure 8c).

As can be observed in Figure 8c, there is a significant difference in the shape of the vertical section of the structure formed from AgNPs from the intensity distribution in the focusing region of the laser radiation. We believe that this can be written off to the synergistic effect of photolytic and thermal effects described above. In this case, several stages of the formation of structures can be distinguished:

(1) At the first stage, exposure to laser radiation leads to the photolytic decomposition of precursor molecules and the formation of a structure with an AgNP concentration distribution close to the distribution of laser radiation intensity. The main difference will be associated with the presence of an intensity threshold below which nanoparticles will not form. According to Figure 8b, the shape of such a structure will be similar to a "filament"—a fairly elongated structure close to the one we previously observed in polymers and porous glasses [21,34]. However, if multiphoton absorption predominates, a more compact "core"

of AgNPs can form in the focusing region. This is due to the fact that in a multiphoton process the result is determined by the Nth power of the laser field intensity, where N > 1.

(2) Absorption and scattering of laser radiation in the formed structure in the form of a "filament" leads to the fact that the intensity in the filament gradually decreases with increasing depth relative to the original one.

(3) The gradual accumulation of AgNPs in the "filament" leads to screening of the lower part of the structure, where laser radiation ceases to flow.

(4) The above processes lead to the fact that the time of laser radiation exposure to AgNPs in the formed structure decreases with the depth of its occurrence. Correspondingly, the time of laser heating of the structures formed from nanoparticles also decreases. Therefore, the shape of the resulting structure is transformed, taking into account both light and thermal effects. As a result, the original "filament" structure grows upward and takes the form of an inverted drop.

So, after all these steps, the cross-sectional view of the AgNP structure formed in Vycor glass acquires a teardrop shape (Figure 5c). As for the MCTU-RF glasses, the shapes of the structures formed in them are more similar to the filament-like distribution of the laser radiation intensity (Figure 8b). We explain this by the lesser role of thermal processes due to the initially lower density of precursor molecules in the bulk of the porous matrix.

It should be underlined that, in the area of formed structures with a high concentration of AgNPs, strong heating of the matrix can lead to melting and deformation of the pore walls of nanoporous glass. Such a transformation will lead to a local change in the refractive index, which is observed in the region of the formed structures after annealing the samples at temperatures above 600 °C (Figure 3b,f).

## 4. Conclusions

It is shown that effective saturation of nanoporous glasses with organometallic silver compound Ag(hfac)COD can be carried out in supercritical $CO_2$. It has been established that the formation of microstructures from AgNPs, including those of submicron size, on the surface and in the volume of impregnated nanoporous quartz glasses can be performed using focused high-frequency femtosecond laser radiation with a low pulse energy $\leq 1$ nJ.

It has been discovered that the differences in the shape of the AgNP microstructures formed in the region of laser radiation focusing are associated with different degrees of porosity and precursor concentration in pores. At the same time, the shape and contrast of microstructures in the volume of nanoporous glasses are associated with the synergistic effect of photolytic and thermal mechanisms. During laser exposure the temperature gradually increases as a result of photolytic decomposition of the precursor due to the absorption of laser radiation on the resulting nanoparticles. This leads to activation of the mechanism of thermal decomposition of the precursor and acceleration of the formation of AgNPs. It is the development of this avalanche-like process that explains the jump-like increase in temperature observed in the experiment some time after switching on the laser radiation. The boundaries of the synergistic effect on temperature heating for these samples were determined, which were 40 °C for Vycor and 50 °C for MCTU-RF porous glasses. It has also been established that the creation of plasmonic structures from AgNPs is accompanied by the appearance of extremely stable heat-resistant "imprints" of the micron scale, associated with local destruction of the glass matrix in this region. The results obtained are of interest for both fundamental and numerous practical applications related to the laser formation of surface and bulk structures from nanoparticles in micro- and nanoporous materials.

**Author Contributions:** Conceptualization, V.I.Y., A.O.R. and V.N.S.; methodology, N.V.M., A.O.R., E.O.E., S.S.F. and N.V.M.; software, V.I.Y. and E.O.E.; validation, V.I.Y. and N.V.M.; formal analysis, V.I.Y.; investigation, V.N.S.; resources, V.N.S., S.S.F. and N.V.M.; data curation, A.O.R. and N.V.M.; writing—original draft preparation, A.O.R., E.O.E. and V.I.Y.; writing—review and editing, V.N.S., S.S.F., V.I.Y. and N.V.M.; visualization, E.O.E., A.O.R. and V.I.Y.; supervision, N.V.M., V.I.Y. and A.O.R.;

project administration, N.V.M. and V.I.Y.; funding acquisition, N.V.M. All authors have read and agreed to the published version of the manuscript.

**Funding:** This research was funded by State assignment of the Federal Research Center "Crystallography and Photonics" and partially supported by by State assignment of Moscow State Univercity funding on the topic: No. 115041410201 "Formation of nanostructural objects and their study by spectroscopic methods".

**Data Availability Statement:** The data presented in this study are available on request from the corresponding author.

**Acknowledgments:** The authors would like to thank Svetlana Perfilyeva for her help in translating the manuscript.

**Conflicts of Interest:** The authors declare no conflict of interest.

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
