# Peer review of "Femtosecond Laser Fabrication of Silver Microstructures in Nanoporous Glasses"

_photonics, doi:10.3390/photonics10091055_

Round 1

Reviewer 1 Report

Thanks for the submission of this interesting manuscript. The work done contribute to understanding the mechanism of Femtosecond laser fabrication of silver microstructures in nanoporous glasses.

1.     I suggest letting the manuscript be reviewed by a native English speaker since some sentences are hard to understand, i.e., ‘In this paper the results of the study of the process of laser formation of microstructures from silver nanoparticles in nanoporous quartz glasses are presented’—very wordy sentence.

2.     I find the results reported a bit too minimalist and poorly explained accurately. You introduce the importance of “femtosecond laser radiation with a wavelength of 525 nm and energy in the pulse up to 1 nJ”. However, it is not clear how did you find a specified wavelength. Please give more details about the used 525 nm wavelength. Have you tried any other wavelength? And if not why

3.     Your experimental process involves several parameters (a pulse repetition rate of 70 MHz, a pulse duration of 200 fs, and an individual pulse energy of 0.95 nJ was fed into the lens through a system of mirrors). It would have been a good idea to explain these parameters and add them to the manuscript to enrich its relevance to the scientific community.

4.     The following reference is highly recommended for the integration of this

Article.

                I.          https://doi.org/10.1007/s10854-022-09549-z

5.     In this paper, in lines 150-151, you used two objectives with different apertures, “can you please tell us the distance between the two objectives, height, etc.?

6.     In Fig. 2(d, e), why are rectangular shapes different if both photographs are the same size?

I suggest letting the manuscript be reviewed by a native English speaker since some sentences are hard to understand, i.e., ‘In this paper the results of the study of the process of laser formation of microstructures from silver nanoparticles in nanoporous quartz glasses are presented’—very wordy sentence.

Author Response

Dear Reviewer,
Thank you for appreciating our article.
Please find below our responses to your questions and suggestions.
1.    I suggest letting the manuscript be reviewed by a native English speaker since some sentences are hard to understand, i.e., ‘In this paper the results of the study of the process of laser formation of microstructures from silver nanoparticles in nanoporous quartz glasses are presented’—very wordy sentence. 
Answer: We thank the reviewer for carefully reading the manuscript. We have taken note of your comment and made the necessary corrections.
2.    I find the results reported a bit too minimalist and poorly explained accurately. You introduce the importance of “femtosecond laser radiation with a wavelength of 525 nm and energy in the pulse up to 1 nJ”. However, it is not clear how did you find a specified wavelength. Please give more details about the used 525 nm wavelength. Have you tried any other wavelength? And if not why
Answer: Thank you for your comment. The wavelength of 525 nm used by us corresponds to the second harmonic from the fundamental generation line (1050 nm) of the Yb3+ fiber laser. This is reported in the article in line 131. Of greatest interest in our studies is the use of femtosecond laser radiation to form silver nanostructures in the matrix. To solve our problem, we chose this type of available femtosecond lasers, since the wavelength in the region of 520-530 nm is well suited for efficient decomposition of the silver precursor, as was shown in our earlier works [10.1016/j.supflu.2018.04.003, fig .2]. We suggest that similar results would be obtained using femtosecond radiation with a different wavelength, but the study of this issue is beyond the scope of this publication and will be considered by us separately in the next work.
3.    Your experimental process involves several parameters (a pulse repetition rate of 70 MHz, a pulse duration of 200 fs, and an individual pulse energy of 0.95 nJ was fed into the lens through a system of mirrors). It would have been a good idea to explain these parameters and add them to the manuscript to enrich its relevance to the scientific community.

Answer: Thanks for the suggestion. As we understand, you are offering to explain why these particular parameters were used. The idea was to study the features of the formation of structures in porous glass using femtosecond radiation exposure to a silver precursor embedded in a matrix. For these studies, one of the available types of laser systems with the indicated parameters was chosen. The repetition rate and pulse duration are primarily set by the configuration of the laser device, varying them with additional blocks is beyond the scope of this work. We agree with you that it is of great interest to compare the obtained results with others. Such a future study is planned by us.

4.     The following reference is highly recommended for the integration of this
Article.
I.    https://doi.org/10.1007/s10854-022-09549-z
II.    https://doi.org/10.1364/AO.484465
Answer: The authors of the article thank the referee for the suggestion to use these links. Article [10.1007/s10854-022-09549-z] presents interesting optical techniques for studying solid materials. In our opinion, this link is not very suitable for the presented article. However, in the next planned publication on this topic, we will try to include it without fail. Another suggested article [10.1364/AO.484465] is also very interesting. This article proposes a new portable polarizing parametric indirect microscopy. We can use this useful manuscript when publishing our results on the effect of laser radiation on biological tissues.
5.     In this paper, in lines 150-151, you used two objectives with different apertures, “can you please tell us the distance between the two objectives, height, etc.? 
Answer: These two objectives were installed manually and used in turn. If you mean the working distances of these objectives, then they are listed in Table 2 of this article.
6.     In Fig. 2(d, e), why are rectangular shapes different if both photographs are the same size? 
Answer: The scale marks shown at the bottom of each of the photographs (2d,e) show approximately 10% smaller scale marks in the case of Fig. 2e. This creates the effect of different rectangular shapes on these images. 
On behalf of authors, Dr. Nikita Minaev.

Reviewer 2 Report

The manuscript entitled “Femtosecond laser fabrication of silver microstructures in nanoporous glasses” is focused on understanding the mechanisms of Ag NPs formation under the irradiation of femtosecond laser in the impregnated porous glass. The mechanism was established. The manuscript is well-written. However, the following issues must be addressed before this paper can be considered for publication in this journal.

 1.       There is a typo in page 2, “The sensitivity of the entire system is ensure by a high degree of pore percolation, which in this case allows the detectable ….”

2.       Please explain the necessity of using a supercritical carbon dioxide (scCO2) medium here for the impregnation of precursor Ag(hfac)COD into a porous matrix.

3.       Page 8: “At the heat treatment temperatures used (~600℃), the initial nanoparticles (less than 4 nm in size) begin to change their appearance due to their melting and evaporation. In this case, it becomes possible to move silver atoms through the pores in the glass with the formation of new larger thermally stable nanoparticles.” Could you please provide the experimental results, such as the SEM or TEM results, to evidence the initial formation of the 4nm Ag NPs (in addition to only refer to the literature)? Also, could you please provide the structural evolution of the Ag NPs as the reaction time goes on to directly prove the hypothesis?

4.       Page 8: “Here, the probability of deformation of the initial pores with a tendency to increase their size under the action of femtosecond radiation is the highest at the maximum beam intensity.” “In our case though, the impact of femtosecond radiation from the viewpoint of pore transformation can be more effective because of the photothermal stimulation processes of AgNPs.” Could you please perform the contrast experiments to show the pore transformation under the femtosecond radiation with and without the Ag NPs to illustrate the effect of photothermal stimulation?

5.       Page 9: “This indicates the formation of spherical AgNPs about 4 nm in size in glass pores. As noted, nanoparticles of this type also appear in impregnated Vycor glasses subjected to the action of continuous laser radiation with a wavelength of 532 nm. Based on the shape of the plasmon absorption spectra (Fig.4) it can be assumed that the shape and size of the synthesized AgNPs in glass pores remain almost unchanged at the given radiation parameters.” Could you please provide the experimental results, such as the SEM or TEM results, to evidence the formation of this 4nm Ag? Also, could you please provide the direct image of the unchanged Ag NPs size to support the proposed mechanism?

Language is not a big issue here

Author Response

Dear Reviewer!
We are grateful for your careful reading of our article, for your questions and valuable comments, which actually initiated our additional analysis of the article.
Please find below our responses to your questions and suggestions
1.       There is a typo in page 2, “The sensitivity of the entire system is ensured by a high degree of pore percolation, which in this case allows the detectable ….”
Answer: We thank you for finding typos. We also eliminated the typos found in the text of the article in the following lines 60, 61, 72, 74, 101, 384.
2.       Please explain the necessity of using a supercritical carbon dioxide (scCO2) medium here for the impregnation of precursor Ag(hfac)COD into a porous matrix.
Answer: The use of supercritical carbon dioxide in the impregnation of organometallic precursors into porous matrices has been practiced by us for more than 20 years (see references to the works of our team of authors). As a solvent, it is environmentally friendly; it is easily removed from the matrix pores upon transition to a gaseous state. Due to its low viscosity in the supercritical state, it is well introduced together with precursor molecules into interconnected nanometer-sized pores. The transformation of carbon dioxide into the supercritical state does not require high temperatures and pressures, which makes this method an accessible on a laboratory scale.
3.       Page 8: “At the heat treatment temperatures used (~600℃), the initial nanoparticles (less than 4 nm in size) begin to change their appearance due to their melting and evaporation. In this case, it becomes possible to move silver atoms through the pores in the glass with the formation of new larger thermally stable nanoparticles.” Could you please provide the experimental results, such as the SEM or TEM results, to evidence the initial formation of the 4nm Ag NPs (in addition to only refer to the literature)? Also, could you please provide the structural evolution of the Ag NPs as the reaction time goes on to directly prove the hypothesis?
Answer: Thank you for your interesting offer. We hope to do similar work in the future. We already have experience and developed special approaches for studying samples using simultaneous optical spectroscopy of plasmon absorption of the TEM technique and studying the transformation of AgNPs in impregnated Vycor glass. Similar transformations were observed by us when exposed to continuous laser radiation of different wavelengths, including those with a close wavelength of 532 nm ([14] - 10.1016/j.supflu.2018.04.003 and 10.1016/j.supflu.2017.03.). We obtained the same results when forming Ag NPs from the same precursor and in polymer matrices, which was shown using TEM - (10.1134/S1995078010070025 and 10.1002/lapl.200910159). More details about the formation of such AgNPs are described in [Klimov, Vasily. Nanoplasmonics. CRC press, 2014]. Of course, the mechanism of formation of AgNPs in porous glasses under the action of femtosecond radiation may differ from that presented in [14] and will require separate consideration with the formation of a large series of experimental samples for measurements using electron microscopy.
4.       Page 8: “Here, the probability of deformation of the initial pores with a tendency to increase their size under the action of femtosecond radiation is the highest at the maximum beam intensity.” “In our case though, the impact of femtosecond radiation from the viewpoint of pore transformation can be more effective because of the photothermal stimulation processes of AgNPs.” Could you please perform the contrast experiments to show the pore transformation under the femtosecond radiation with and without the Ag NPs to illustrate the effect of photothermal stimulation?
Answer: To carry out experiments demonstrating the possibility of deformation of glass pores in regions where AgNPs are created under the action of femtosecond laser radiation, it is necessary to use a special technique. This technique is currently under development and will include the use of an ion beam to cut thin micron cross-sections of porous glass samples with structures formed from AgNPs, followed by TEM analysis.
5.       Page 9: “This indicates the formation of spherical AgNPs about 4 nm in size in glass pores. As noted, nanoparticles of this type also appear in impregnated Vycor glasses subjected to the action of continuous laser radiation with a wavelength of 532 nm. Based on the shape of the plasmon absorption spectra (Fig.4) it can be assumed that the shape and size of the synthesized AgNPs in glass pores remain almost unchanged at the given radiation parameters.” Could you please provide the experimental results, such as the SEM or TEM results, to evidence the formation of this 4nm Ag? Also, could you please provide the direct image of the unchanged Ag NPs size to support the proposed mechanism?
Answer: At present, we plan to carry out experimental work on the direct determination of the size and shape of AgNPs in porous glasses with a silver precursor under the action of femtosecond laser radiation. As we mentioned above, answering your question No. 3, the process of directly measuring the size and shape of nanoparticles under these conditions is a rather complex technological task (see our work at the link [14 - 10.1016/j.supflu.2018.04.003]. It is necessary to prepare a sufficient number of samples of porous glasses irradiated for different times, and then carefully break each sample of irradiated glass to prepare measurements for TEM, which is a long and complicated procedure. In the cited work, the situation with the production of nanoparticles in glass under the action of continuous laser radiation with a wavelength of 532 nm was considered. The obtained results on the sizes of nanoparticles correspond to the data of the analysis of the corresponding plasmon absorption spectra. A similar analysis of the shape and position of the maxima of these spectra was also carried out in our other works (see references [17] [18] from the same list and the article [10.3103/S0027134914030035]) and corresponded to spherical particles about 4 nm in size. Also, we are developing a technique for time-resolved measurements of the plasmon resonance absorption spectra of formed silver nanoparticles using a light source based on a femtosecond supercontinuum. As for the last wish, as shown by the results presented in this work, with an increase in the irradiation dose, the appearance of aggregation of nanoparticles can be expected. This can make it difficult to analyze their shape as the laser radiation dose increases.
On behalf of authors, Dr. Nikita Minaev.

Reviewer 3 Report

This detailed study investigates the effects of femtosecond laser irradiation-induced formation of AgNPs microstructures on the surface and in the volume of porous quartz glasses under different irradiation conditions. The work is highly professional and detailed, providing a comprehensive overview of the parameters that affect the abovementioned process.

I have two minor suggestions that could help clarify some points in the text:

I belive it would be beneficial to include few explanations regarding the (non-linear) absorption behavior of the nanoparticles during the exposure process and how this may impact the overall procedure. For example, the mention of the multiphoton absorption and the connection with the laser-induced decomposition (in lines 40-43), or in the case were you distiquished several stages of the structures' formation (at 1 and 2).

In Figure 4, it may be clearer to indicate that the purple spectrum represents the Vycor glass prior to exposure.

Author Response

Dear Reviewer,
Thank you for appreciating our article.
Please find below our responses to your questions and suggestions.
1. I belive it would be beneficial to include few explanations regarding the (non-linear) absorption behavior of the nanoparticles during the exposure process and how this may impact the overall procedure. For example, the mention of the multiphoton absorption and the connection with the laser-induced decomposition (in lines 40-43), or in the case were you distiquished several stages of 
Answer: Thank you for your valuable suggestions. The corresponding sentences (lines 40-43) and (lines 569-569) have been rewritten and supplemented. As for the detailed discussion of the questions on the degree of linearity of absorption processes both by the nanoparticles themselves and in the case of photoinduced decomposition of precursor molecules, we decided not to consider them in this article because of its large size. We will touch on these really interesting questions in our subsequent publications, since they require the development of separate approaches and new experiments, including the development of a new technique for time-resolved diagnostics of the AgNPs formation process.
2. In Figure 4, it may be clearer to indicate that the purple spectrum represents the Vycor glass prior to exposure.
Answer: We thank the referee for useful comments. We have added the explanation "The purple spectrum (5) represents the pure Vycor glass without precursor and prior to exposure" to the caption to Fig. 4.
On behalf of authors, Dr. Nikita Minaev.

Round 2

Reviewer 2 Report

I can recommend the publication of this paper in its present form